# Impact of Age and Comorbidity on Multimodal Management and Survival from Colorectal Cancer: A Population-Based Study

**DOI:** 10.3390/jcm10081751

**Published:** 2021-04-17

**Authors:** Ilmo Kellokumpu, Matti Kairaluoma, Jukka-Pekka Mecklin, Henrik Kellokumpu, Ville Väyrynen, Erkki-Ville Wirta, Eero Sihvo, Teijo Kuopio, Toni T. Seppälä

**Affiliations:** 1Department of Gastrointestinal Surgery, Central Finland Hospital Nova, 40620 Jyväskylä, Finland; ilmo.kellokumpu@fimnet.fi (I.K.); matti.kairaluoma@ksshp.fi (M.K.); jukka-pekka.mecklin@ksshp.fi (J.-P.M.); ville.vayrynen@ksshp.fi (V.V.); 2Faculty of Sports and Health Sciences, University of Jyväskylä, 40014 Jyväskylä, Finland; 3Department of Internal Medicine, Turku University Hospital, 20521 Turku, Finland; henrik.kellokumpu@gmail.com; 4Department of Gastrointestinal Surgery, Tampere University Hospital, 33521 Tampere, Finland; erkki-ville.wirta@fimnet.fi; 5Department of Thoracic Surgery, Central Finland Hospital Nova, 40620 Jyväskylä, Finland; eero.sihvo@ksshp.fi; 6Department of Pathology, Central Finland Hospital Nova, 40620 Jyväskylä, Finland; teijo.kuopio@ksshp.fi; 7Department Biological and Environmental Science, University of Jyväskylä, 40014 Jyväskylä, Finland; 8Department of Gastrointestinal Surgery, Helsinki University Hospital, 00290 Helsinki, Finland; 9Department of Surgical Oncology, Johns Hopkins Hospital, Baltimore, MD 21287, USA; 10iCAN Precision Medicine Flagship, Helsinki University, 00290 Helsinki, Finland

**Keywords:** elderly, comorbidity, colorectal cancer, survival

## Abstract

This retrospective population-based study examined the impact of age and comorbidity burden on multimodal management and survival from colorectal cancer (CRC). From 2000 to 2015, 1479 consecutive patients, who underwent surgical resection for CRC, were reviewed for age-adjusted Charlson comorbidity index (ACCI) including 19 well-defined weighted comorbidities. The impact of ACCI on multimodal management and survival was compared between low (score 0–2), intermediate (score 3) and high ACCI (score ≥ 4) groups. Changes in treatment from 2000 to 2015 were seen next to a major increase of laparoscopic surgery, increased use of adjuvant chemotherapy and an intensified treatment of metastatic disease. Patients with a high ACCI score were, by definition, older and had higher comorbidity. Major elective and emergency resections for colon carcinoma were evenly performed between the ACCI groups, as were laparoscopic and open resections. (Chemo)radiotherapy for rectal carcinoma was less frequently used, and a higher rate of local excisions, and consequently lower rate of major elective resections, was performed in the high ACCI group. Adjuvant chemotherapy and metastasectomy were less frequently used in the ACCI high group. Overall and cancer-specific survival from stage I-III CRC remained stable over time, but survival from stage IV improved. However, the 5-year overall survival from stage I–IV colon and rectal carcinoma was worse in the high ACCI group compared to the low ACCI group. Five-year cancer-specific and disease-free survival rates did not differ significantly by the ACCI. Cox proportional hazard analysis showed that high ACCI was an independent predictor of poor overall survival (*p* < 0.001). Our results show that despite improvements in multimodal management over time, old age and high comorbidity burden affect the use of adjuvant chemotherapy, preoperative (chemo)radiotherapy and management of metastatic disease, and worsen overall survival from CRC.

## 1. Introduction

Colorectal carcinoma (CRC) is the third most common cancer worldwide [1,2] and the fourth leading cause of cancer death [1,2]. Over the years, many improvements have been made in the management of colorectal cancer [3,4,5,6,7,8]. These include diagnostic procedures, use of laparoscopic surgery, improvements in perioperative care, refinement of histopathological examination and the development of effective neoadjuvant, adjuvant and palliative treatments and extended indications for resection of metastatic disease.

CRC incidence increases markedly with increasing age as do the prevalence of chronic comorbid conditions [1]. Due to frequent medical comorbidities and aging-related diminished physiological reserves, elderly patients are considered at higher risk for complications from major cancer surgery and chemotherapy and, therefore increase the complexity of multimodal management of colorectal cancer. In particular, comorbidity may lead to altered treatment, higher morbidity rates and worse survival [9,10,11].

The age-adjusted Charlson comorbidity index (ACCI) is a measure of comorbidity used to standardize the evaluation of surgical patients [12,13,14], and has been reported to be an appropriate prognostic factor for pancreatic, gastric and colorectal cancer patients [15,16,17,18,19,20,21,22]. The age-adjusted Charlson comorbidity index includes 19 well-defined comorbidities and age as an additional factor [14]. The objectives of the current study were to assess the impact of age and comorbidity burden (ACCI) on multimodal treatment and survival from CRC by retrospectively reviewing all diagnosed colorectal cancers from 2000 to 2015 in Central Finland.

## 2. Materials and Methods

According to Finnish healthcare policy, all municipalities are responsible for arranging specialized hospital care for their residents. Each hospital district organizes and provides specialized hospital care for the population in its area. The Central Hospital of Central Finland is the only gastroenterological surgery unit in the Central Finland hospital district. All patients with primary and metastatic colorectal cancer are managed in this hospital, with no referrals to other hospitals. The annual population of the area, obtained from Statistics Finland, averaged around 270,000 during the study period from 2000 to 2015.

### 2.1. Patients

Patients diagnosed with primary colorectal adenocarcinoma from 2000 to 2015 were identified using the histopathological registry of the hospital. Included in the study were patients who underwent resectional surgery for primary colorectal adenocarcinomas, defined as removal of primary tumor. Right-sided colon cancers were defined as those arising from the cecum to, and including, the transverse colon. Left-sided colon cancers were defined as those arising from the splenic flexure down to, and including, the rectosigmoid junction. Tumors located 15 cm or less from the anal verge were considered rectal cancers. 

Colonoscopy, thoracoabdominal computed tomography (CT), endorectal ultrasonography and pelvic magnetic resonance imaging (MRI) were used to diagnose and stage primary colorectal tumors. All patients with colorectal cancer were discussed in multidisciplinary team (MDT) meetings before definitive treatment decisions were made.

### 2.2. Surgical Procedures

Multimodal management was done according to international guidelines [3,4,5,6,7,8]. Significant co-morbidity, inadequate physical and mental performance status and extensive metastatic disease were contraindications for surgery. Patients who were not resected had non-resectional procedures that were purely diagnostic or symptom-alleviating (e.g., stoma), were excluded.

Laparoscopic surgery for colorectal cancer was implemented in 2001 after an initial experience of some 100 laparoscopic colorectal procedures for benign diseases since 1993. Details of our surgical technique, including right or left hemicolectomies, extended hemicolectomies or sigmoid resections with wide mesocolic excision [23,24] and rectal resections according to total mesorectal excision (TME) principles [25], have been described earlier [26,27].

Cancer invasion to uncommon sites including urinary bladder, uterus, ovary or abdominal wall and some emergency situations were treated with extended surgery, when appropriate. Laparoscopic transanal total mesorectal excision (TaTME) was experimentally undertaken in a few patients during 2010–2015. Local endoscopic or transanal excisions for malignant polyps were considered radical for stage I tumors alone. Surgery for metastatic disease was performed when appropriate, according to local MDT. Tumors were staged by staff pathologists according to the TNM/UICC (The Union for International Cancer Control) classification [28]. The quality of the pathological examination was refined since 2005.

### 2.3. Neoadjuvant and Adjuvant Treatments

Adjuvant postoperative chemotherapy for 6 months, consisting of 5-fluorouracil (5-FU) and folic acid, a combination of oxaliplatin with fluorouracil (FOLFOX) or capecitabine (CAPOX) or capecitabine alone for elderly patients, was prescribed to medically fit patients with stage III tumors or high-risk stage II disease. Patients with locally advanced rectal cancer on endorectal ultrasound and magnetic resonance imaging received either a short-course radiotherapy (RT) (5 Gy × 5) followed by surgery within a week or a long-course chemoradiation (1.8 Gy × 25 or 2 Gy × 25 with capecitabine) followed by surgery after 6–8 weeks. A short-course radiotherapy combined with neoadjuvant chemotherapy (FOLFOX or CAPOX) and delayed surgery was used selectively between 2005–2015 for rectal cancer (8). Patients with liver metastases received perioperative chemotherapy with the FOLFOX regimen, with or without biologicals, according to the decision taken at the MDT meeting.

### 2.4. Follow-Up

Follow-up after surgery for primary tumors included initially carcinoembryonic antigen estimation, clinical examination, ultrasound investigation of the liver, and chest radiography every 6 months during the first 3 years, and annually thereafter. Since 2005 chest radiography and ultrasound investigation were replaced by CT. Further characterization of metastases was undertaken by MRI, and after 2005 also by CT–PET (positron emission tomography). Locally recurrent disease was assessed by endoscopy, CT/pelvic MRI and endoscopy. Local recurrence from colon cancer included all peritoneal and/or lymph node metastases in the abdomen. Local recurrence from rectal cancer included recurrences in the small pelvis. All patients included in this study were followed up from time of diagnosis until death or until November, 2020. The date and cause of death were obtained from the medical records and verified from the National Cause of Death Registry.

### 2.5. Data Collection, Assessment of Age-Adjusted Charlson Comorbidity Index and Outcome Measures

Clinical and histopathological data, as well as recurrence data, were retrieved retrospectively from electronic hospital records. The study variables for primary tumors included age, sex, Charlson comorbidity, date of diagnosis, tumor sidedness and location, UICC stage, date and type of surgery, radiotherapy, oncological treatment and survival. Trends in multimodal treatment and survival were examined between 2000–2005, 2006–2010 and 2011–2015. ACCI score was used to quantify weighted comorbidities as summary measures including 19 well-defined comorbidities and age as one additional factor (Table 1) [12,13,14]. The final score was calculated for each patient by taking into account all comorbid conditions present with the exclusion of present colorectal cancer was. Median ACCI score of all CRC patients was 3 (interquartile range (IQR) 2–4). The impact of ACCI on multimodal management and survival was compared between low (score 0–2), intermediate (score 3) and high ACCI (score ≥4) groups. Analyses were performed for age, gender, comorbidity, sidedness (right vs. left) and location (right vs. left vs. rectum), stage distribution, multimodal treatment, local recurrence, metastatic disease and survival. The study was approved by the hospital administrative and ethics board (Dnro13U/2011 and 1/2016 and the National Authority for Welfare and Health (Valvira) (Dnro 3916/06.01.03.01/2016).

### 2.6. Statistical Analysis

Results are given as mean (standard deviation (s.d.)) or median (IQR) values. Pearson’s χ2 or Fisher’s exact tests were used to compare frequencies, analysis of variance (ANOVA) test for comparison of mean values between groups, and Kruskal–Wallis test for comparison of median values between independent samples. The Kaplan–Meier method was used to calculate survival, and differences between groups were compared with the log rank test. Survival times were calculated from the date of primary surgery to the date of death or the end of follow-up November, 2020. Overall survival (OS) included all causes of death. Cancer-specific survival (CSS) included deaths from CRC or its treatment. Disease-free survival was calculated from the date of surgery to the first record of metastasis or local recurrence, and censored to death or to end of follow-up without recurrence. As the number of patients with rectal cancer and a complete pathological response after chemoradiotherapy was small, these patients were included with stage I patients for purposes of survival analyses. Factors affecting survival were analyzed with univariate and multivariable Cox proportional hazards regression models; only variables with *p* < 0.20 were entered in the multivariate analysis. All statistical tests were two-sided. *p* <0.050 was considered significant. STATA^®^ release 13 (Stata Corp, College Station, TX, USA) was used for statistical analysis.

## 3. Results

Of the 1680 CRC patients referred to the hospital between 2000–2015, 1479 underwent resective surgery in the multimodal setting, whereas 201 primary cancers (12.0%) were not operated on, either because of significant co-morbidity, old age and poor performance status, locally unresectable primary cancer, unresectability of metastases or patient’s preference.

### 3.1. Patient Characteristics and Tumor Stage Distribution According to Age-Adjusted Charlson Comorbidity Index (ACCI) Score

The proportion of colon cancers, particularly right-sided cancers, increased over time compared to rectal cancers. Baseline patient characteristics and UICC stage distribution of colon (*p* = 0.044) and rectal (*p* = 0.126) carcinoma according to ACCI groups are presented in Table 2. The incidence of right-sided colon cancers, medical comorbidities and older age was most frequent in the high ACCI group.

### 3.2. Multimodal Treatment 

Changes in treatment over time were seen next to a major increase of laparoscopic surgery, increased use of adjuvant chemotherapy and an intensified treatment of metastatic disease with a shift to an increased use of neoadjuvant chemotherapy and metastasectomy, particularly for rectal carcinoma (Appendix A). Surgical treatment of colon cancer patients with more severe comorbidities increased significantly over time (37.5% vs. 51.6%, *p* < 0.001). A similar time trend was not seen in rectal cancer patients. 

The influence of ACCI on multimodal treatment is presented in Figure 1A,B. Major elective, emergency and local resections for colon carcinoma were evenly performed between the three ACCI groups as were laparoscopic and open resections. (Chemo)radiotherapy for rectal carcinoma was less frequently used, and a higher rate of local excisions and consequently a lower rate of major elective resections was performed in the high ACCI group. Thirty-day mortality after surgery for colon carcinoma in the low, intermediate and high ACCI groups was 0.6%, 0.4% and 6.0%, *p* < 0.001, and 30-day morbidity 18.0%, 22.2% and 31.4%, respectively, *p* < 0.001. The respective figures for rectal cancer were not significantly different: 30-day mortality by the low, intermediate and high ACCI was 1.0%, 0.7% and 3.0%, *p* = 0.190, and 30-day morbidity 39.7%, 34.3% and 35.3%, respectively, *p* = 0.528.

The administration of adjuvant chemotherapy for stages II–III colon carcinoma was less frequent in the high ACCI group. As for rectal cancer, administration of adjuvant chemotherapy was reduced for stage II in the high ACCI group but not for stage III.

Local recurrences after surgery for stages I–III colon and rectal carcinoma, excluding patients who died within 30-days postoperatively, were evenly distributed in the ACCI groups (Figure 1A,B). The incidence of metastatic disease, observed in some 30% of patients over time, was lower in the high ACCI group for colon cancer but not for rectal cancer. Despite increased use of neoadjuvant chemotherapy and metastatic surgery over time, metastatic patients in the high ACCI group received metastatic surgery less often than other patients.

### 3.3. Long-Term Survival

The median follow-up time for patients after resection for colon carcinoma was 6.4 years (IQR 2.3–10.6) years and for rectal carcinoma 7.0 (IQR 3.3–11.4) years. Of the deaths that occurred during the follow-up, non-cancer-related mortality for colon cancer was 30.6% and cancer progression-related mortality 28.8%. Non-cancer-related mortality for rectal cancer was 30.3% and cancer progression-related mortality 25.8%. 

Five-year OS from stages I–IV colon and rectal carcinoma was worse in the high ACCI group compared to low and medium ACCI groups: For colon (Figure 2A), the OS was 71.5% for low ACCI, 71.1% for intermediate ACCI and 49.8% for high ACCI. For rectum (Figure 2B) the OS was 74.4%, 67.8% and 60.4%, respectively. Five-year CSS by low, intermediate and high ACCI from stages I–IV colon was 75.9%, 77.2% and 68.9% (*p* = 0.056) and from rectal cancer 78.3%, 79.1% and 79.4% (*p* = 0.913), respectively. Five-year disease-free survival (DFS) by low, intermediate and high ACCI from stages I–III colon cancer was 78.3%, 80.5% and 80.1% (*p* = 0.904) and from stages I–III rectal cancer 75.9%, 74.4 and 76.5 (*p* = 0.788), respectively.

In multivariable Cox proportional hazard analysis (Table 3) ACCI, male gender, UICC stage and type of surgery at presentation were independent prognostic factors for colon carcinoma and UICC stage and ACCI for rectal carcinoma overall survival.

## 4. Discussion

This retrospective population-based study aimed to examine whether or not the combined effect of age and comorbidity, assessed by the age-adjusted Charlson comorbidity index, had a clinical impact on multimodal management and survival from colorectal cancer. The main findings of the present study were that preoperative radiotherapy, adjuvant chemotherapy, postoperative mortality and morbidity, management of metastatic disease and overall survival were adversely influenced by old age and high Charlson comorbidity burden despite improvements in multimodal therapy and management of metastatic disease over time.

Due to population aging, an increasing proportion of elderly patients and comorbid conditions was observed over time in our study, particularly for colon cancer. The proportion of elderly patients and comorbidity was in our series consistent with other studies [17,18,19,29]. A preponderance of right-sided colon cancers was observed here in line with international trends. About one third of colorectal cancers originated in the rectum, and the rates were higher in men than in women. Proportional UICC stage distribution was relatively evenly distributed between different age-adjusted comorbidity groups.

Changes in treatment from 2000 to 2015 were seen next to a major increase of laparoscopic surgery, increased use of adjuvant chemotherapy and an intensified treatment of metastatic disease with a shift to an increased use of neoadjuvant chemotherapy and metastasectomy, particularly for rectal carcinoma. Laparoscopic and open resections were evenly distributed between the ACCI groups. 

There are interactions between general health status and indications for and tolerability of neoadjuvant and adjuvant treatments, especially in elderly patients. Elderly patients with CRC tend to be underepresented in clinical trials and undertreated in clinical practice. Therefore, the extent to which patients older than 75 years benefit from postsurgical chemotherapy remains a challenge that is frequently encountered in oncology practice [30]. The use of postoperative systemic therapy is generally recommended for Stage III and high risk Stage II colon cancer, but no randomized trials have demonstrated similar gains in the younger and elderly population [31,32]. In addition, the use of postoperative chemotherapy in patients with rectal cancer receiving preoperative radio(chemo)therapy is not based on strong scientific evidence [4,8]. Although the administration of adjuvant chemotherapy increased here during more recent time periods, it was less frequently used in the high ACCI group. A similar trend has been reported by others [11,32].

The use of preoperative (chemo)radiotherapy for rectal carcinoma has not demonstrated an overall survival benefit in randomized trials, but induces tumor regression with a possible complete response (pCR) and reduced local recurrence rate [8]. The administration of preoperative (chemo)radiotherapy for locally advanced rectal cancer was in our series constant between 2000–2015 but was reduced in the high ACCI group. The optimal radiotherapy fractionation and interval between radiotherapy and surgery is still under debate [8]. The Stockholm III trial has shown that delaying surgery for 4–8 weeks after short-course RT, rather than operating immediately, may reduce postoperative complications significantly. The short-course radiotherapy combined with neoadjuvant chemotherapy and delayed surgery was used here selectively since 2008 for rectal cancer.

Indicators of quality of treatment, e.g., proportion of patients operated in an emergency setting, risk of postoperative mortality and complications and locoregional recurrence were in agreement with the findings from previous studies. The rate of surgical emergency, usually bowel obstruction and/or perforation, resulting in higher mortality, increased local recurrence rate and worse survival, was in our series 10% in contrast to 17–29% of patients reported internationally [33]. Postoperative mortality rate in this patient series, including also emergency surgery, was low and within the reported range of 2–5% from other European countries [34,35,36]. The increase in postoperative mortality with advancing age may be partly explained by comorbidity. The overall short-term morbidity was in line with other reports [36], but was higher especially among older patients with high comorbidity. Local recurrence rate after colon and rectal surgery remained relatively stable over time, and was in agreement with the figures reported earlier: colon cancer 10% and rectal cancer 5–15% [29,37].

Metastatic disease in patients having undergone surgical resection for primary colorectal cancer was in our series constant (about 30%) throughout the study period, and was more aggressively managed over time by neoadjuvant chemotherapy and metastatic surgery. Overall, the metastasectomy rate for CRC metastases compares favorably with those of 12% to 30% reported in other population-based studies [38,39,40] despite frequent multisite metastatic patterns. Liver and pulmonary metastasectomy are generally recommended for highly selected subsets of elderly patients due to concerns about the operation safety and shorter life expectancy. In our series the use metastasectomy was significantly reduced in the high ACCI group due to patient comorbidities, high age and multisite metastatic patterns. Metastatic surgery consisted here mostly of liver and lung resections, being increasingly performed using laparoscopic and thoracoscopic approach. Bowel and other organ resections were occasionally performed.

OS showed a significant difference in favor of the younger and more fit patients whereas no differences were observed in CSS and DFS. The difference in OS between the ACCI groups persisted over time and across all UICC stages. Our results are in line with previous cohort studies that did find an inverse relationship between age, comorbidity and survival [9,10,11]. However, the observation that cancer-related mortality does not decrease with increasing age strengthens the idea that, although elderly patients have a shorter life expectancy based on their age and pre-existent conditions, they do still benefit from cancer treatment. It seems that patients with high ACCI present similar oncological results despite being treated less intensively in terms of chemotherapy and multimodal treatment.

The results herein reported should be interpreted with some caution, however. First, the number of patients is relatively small compared to large nationwide population-based studies. A considerable number of high-risk patients in our study were not operated on at all, which may contribute to acceptable complications and mortality numbers. Pre-operative performance and nutritional status, cachexia and the presence of multisite metastases may have contributed to the selection of these patients. Second, preoperative chemoradiotherapy in rectal cancer might have shifted stage-specific outcome, as postoperative stage has been used in this study. Patients who respond well to preoperative treatment have been downstaged, thereby deteriorating survival rates in the higher stages. Third, for modern treatment of CRC, routine testing of microsatellite instability (MSI) and *RAS/BRAF* mutation status has been recommended since 2014, as these have been confirmed to be prognostic factors for CRC patients. Our group has previously reported MSI and *RAS/BRAF* status as well as immune cell score in a subset of this patient series [41,42] and, therefore, MSI and *RAS/BRAF* status as well as detailed histopathological tumor characteristics were not addressed here. Primary tumor sidedness has also been emphasized in recent years in recognition of the fact that the side of origin plays a role in tumor behavior and progression [43]. Tumors originating in the right colon are more frequently associated with female patients, the elderly, *BRAF* mutations, the enhanced CpG island methylator phenotype, high microsatellite instability, and high expression of consensus molecular subtypes 1 and 3 compared with left-side origin tumors. However, in our series there was no significant survival difference between right- and left-sided colon carcinomas. The strengths of our study are its population-based design, reliable cancer recurrence data and detailed follow-up of all patients using direct methods (medical chart review) and national Death Registry data, thus providing a real-life data of the multimodal management of CRC and potential results of therapy at the population level.

Despite improvements in multimodal management over time, old age and high comorbidity burden affect the use of adjuvant chemotherapy, preoperative (chemo)radiotherapy and management of metastatic disease and worsen overall survival from CRC.

## Figures and Tables

**Figure 1 jcm-10-01751-f001:**
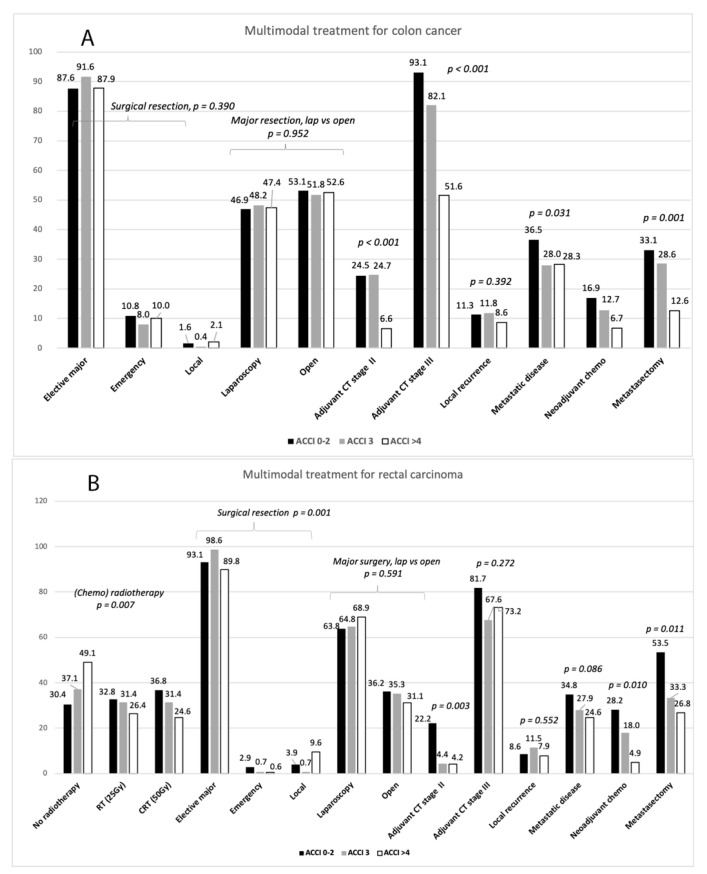
Multimodal management for colon (**A**) and rectal (**B**) carcinoma by low, intermediate and high ACCI groups. Bars represent n (%) except *p*-values.

**Figure 2 jcm-10-01751-f002:**
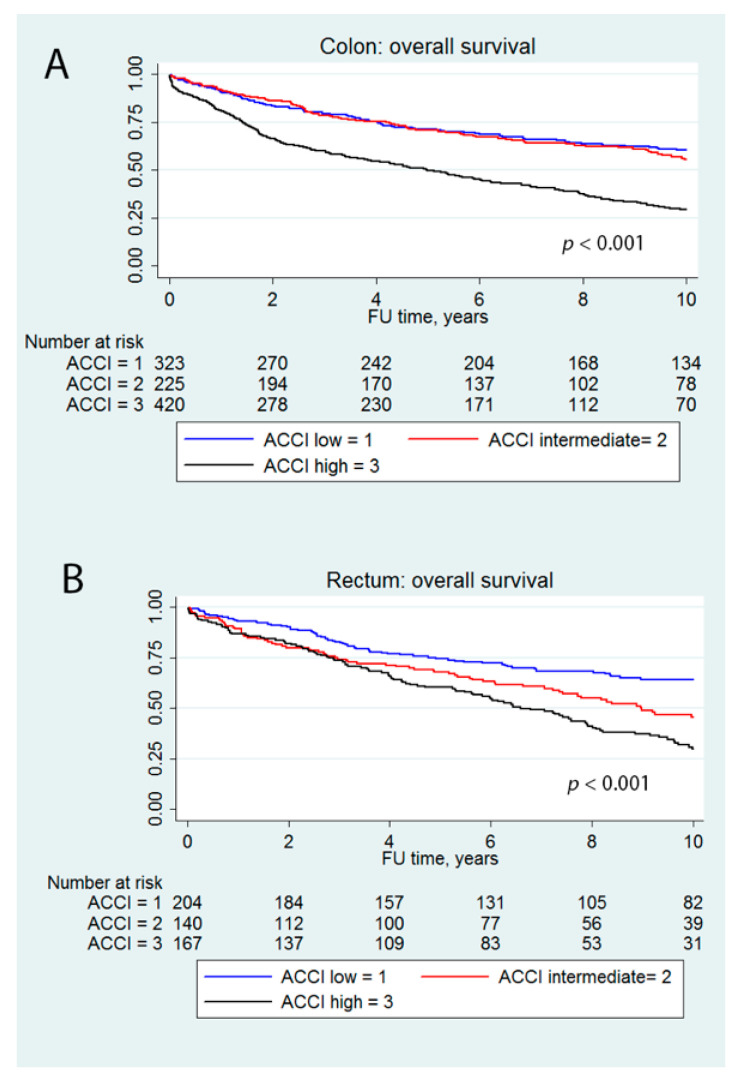
Five-year overall from stages I–IV colon carcinoma (**A**) and rectal carcinoma (**B**).

**Table 1 jcm-10-01751-t001:** Age-adjusted Charlson comorbidity index and comorbidity weighting.

Weight	Charlson Comorbid Condition
1	Myocardial infarction, congestive heart failure, peripheral or cerebral vascular disease, TIA (transient ischemic attack), chronic obstructive pulmonary disease (COPD), connective tissue disease, mild liver disease, peptic ulcer disease, diabetes, dementia
2	Hemiplegia, moderate/severe renal disease, diabetes with end-organ damage, previous cancer *, leukemia, lymphoma
3	Moderate or severe liver disease
6	Metastatic solid cancer *, Acquired Immunodeficiency Syndrome (AIDS)
1	For each decade over age 50 years, up to 4 points

* if treated previously before the diagnosis of present colorectal cancer.

**Table 2 jcm-10-01751-t002:** Patient characteristics and tumor stage distribution by low, intermediate and high age-adjusted Charlson comorbidity index (ACCI).

	All Patients	ACCI = 0–2 Low	ACCI = 3 Intermediate	ACCI = ≥4 High	*p*-Value
Primary tumour site Right colon Left colon Rectum	558 (37.7) 410 (27.7) 511 (34.6)	171 (32.3) 152 (28.7) 207 (39.1)	124 (34.2) 101 (27.8) 138 (38.0)	263 (44.9) 157 (26.8) 166 (28.3)	<0.001
*Colon* Age (years), mean (SD)	968 (100.0) 70.7 (11.3)	323 (100.0) 59.1 (8.1)	225 (100.0) 71.3 (6.2)	420 (100.0) 79.4 (6.8)	<0.001
Gender, male	471 (48.7)	155 (48.0)	108 (48.0)	208 (49.5)	0.894
Charlson comorbidity	412 (42.6)	19 (5.9)	55 (24.4)	338 (80.5)	<0.001
Disease stage (UICC) I II III IV	168 (17.4) 365 (37.7) 304 (31.4) 131 (13.5)	58 (18.0) 106 (32.8) 102 (31.6) 57 (17.6)	32 (14.2) 93 (41.3) 78 (34.7) 22 (9.8)	79 (18.6) 166 (39.5) 124 (29.5) 52 (12.4)	0.044
*Rectum* Age(years), mean (SD)	511 (100.0) 67.9 (10.7)	204 (100.0) 59.0 (7.6)	140 (100.0) 70.4 (7.0)	167 (100.0) 76.7 (7.8)	<0.001
Gender male	331 (64.8)	134 (65.7)	94 (67.1)	103 (61.7)	0.571
Charlson comorbidity	185 (36.2)	9 (4.4)	49 (35.0)	127 (76.1)	<0.001
Disease stage (UICC)					
0 (pCR) ^a^ I II III IV	10 (2.0) 170 (33.3) 147 (28.8) 138 (27.0) 46 (9.0)	3 (1.5) 60 (29.4) 54 (26.5) 63 (29.4) 27 (13.2)	4 (2.9) 46 (32.9) 45 (32.4) 37 (26.4) 8 (5.7)	3 (1.8) 64 (38.2) 48 (28.7) 41 (24.6) 11 (6.6)	0.168

Figure are n (%) unless otherwise stated; ^a^ pCR = pathologic complete response.

**Table 3 jcm-10-01751-t003:** Univariate and multivariable Cox proportional hazard analysis for overall survival.

	Univariate Hazard Ratio (HR) (95% Confidence Interval (CI))	*p*-Value *	Multivariable HR	*p*-Value *
			(95% CI)	
*Colon*				
Age-adjusted comorbidity index score (ACCI)				
Low	1	<0.001	1	<0.001
Intermediate	1.30 (1.02–1.66)		1.50 (1.17–1.92)	
High	2.70 (2.21–3.30)		3.24 (2.65–3.98)	
Gender				
Male	1	0.032	1	0.026
Female	0.84 (0.71–0.98)		0.82 (0.69–0.97)	
UICC stage				
I	1	<0.001	1	<0.001
II	1.25 (0.97–1.63)		1.31 (1.00–1.73)	
III	1.53 (1.18–1.20)		1.71 (1.29–2.26)	
IV	7.34 (5.47–9.83 )		8.75 (6.38–11.99)	
Type of surgical resection				
Elective major surgery	1	<0.001	1	<0.001
Emergency surgery	2.21 (1.72–2.83)		1.82 (1.41–2.35)	
Local excision	1.36 (0.75–2.48)		1.90 (1.0–3.62)	
Adjuvant chemotherapy				
No	1	0.254	_	
Yes	0.90 (0.76–1.07)			
Side of colon				
Right	1	0.141	1	0.077
Left	0.88 (0.75–1.04)		0.86 (0.72–1.02)	
Time periods				
2000–2005	1	0.707		
2006–2010	0.96 (0.79–1.17)		_	
2011–2015	0.96 (0.77–1.19)			
*Rectum*				
ACCI				
Low	1	<0.001	1	<0.001
Intermediate	1.81 (1.33–2.45)		2.25 (1.65–3.07)	
High	2.51 (1.89–3.34)		3.29 (2.42–4.36)	
Gender				
Male	1	0.320	_	
Female	0.88 (0.69–1.13)			
UICC stage				
I	1	<0.001	1	<0.001
II	1.33 (0.98–1.80)		1.41 (1.04–1.91)	
III	1.59 (1.16–2.16)		1.90 (1.39–2.60)	
IV	5.51 (3.73–8.15)		8.42 (5.60–12.67)	
Type of surgical resection				
Elective major surgery	1	0.543	_	
Emergency surgery	3.11 (1.46–6.60)			
Local excision	1.01 (0.58–1.76)			
Preoperative radiotherapy				
No radiotherapy	1	0.296	_	
Short-course (25 Gy)	0.87 (0.65–1.16)			
Chemoradiotherapy (50 Gy)	1.17 (0.89–1.55)			
Adjuvant chemotherapy				
No	1	0.601	_	
Yes	1.07 (0.83–1.39)			
Time periods				
2000–2005	1	0.483	_	
2006–2010	1.01 (0.76–1.34)			
2011–2015	0.89 (0.66–1.22)			

* *p* for linearity.

## Data Availability

The data are available from authors by a reasonable request.

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
