# Peer review of "Impact of Age and Comorbidity on Multimodal Management and Survival from Colorectal Cancer: A Population-Based Study"

_jcm, 2021, doi:10.3390/jcm10081751_

Round 1
Reviewer 1 Report
Thanks for letting me review this interesting work. This work by Kellokumpu et al. examined the impact of time trend, age and comorbidity burden on multimodal management and survival among CRC patients. Changes in treatment from 2000 to 2015 were seen next to a major increase of laparoscopic surgery, increased use of adjuvant chemotherapy and an intensified treatment of metastatic disease. Overall and cancer-specific survival from stage I-III CRC remained stable over time, but survival from stage IV improved. The five-year overall survival but not five-year cancer-specific and disease-free survival rates from colon and rectal carcinoma was worse in the high ACCI group compared to low ACCI group. Cox model showed that high ACCI was an independent predictor of poor overall survival. The strengths of this study include the use of reliable cancer recurrence data and detailed follow-up of all patients using medical chart review and national Death Registry data. The limitations include small sample size, potential selection bias due to a portion of patients not operated on, and altered cancer stage from pre-operative to post-operative. Below are more specific comments:
Comments
I think the entire manuscript contains an overwhelming amount of data. The descriptive statistics in table 2, table 3, and supplementary table 1 are especially distracting. I think many readers will get lost in the process of reading. I would recommend showing only the more important data that support the study hypothesis and keep the paragraphs on descriptive statistics concise. The authors can also make the subheadings bold in the tables to facilitate reading.
Time trend and ACCI are two important layers in this study but the mixture of the two factors may confuse the readers as to what the focus of this paper really is. I think the authors should focus on one of them in this study.
Figure 2, figure 3, and table 4 to me are more interesting and should be the “main analysis”. The manuscript will benefit from focusing on these tables/figures. The descriptive statistics should probably be moved to the supplemental section and the authors can just briefly mention them in a small paragraph.
Why did the age for colon cancer patients increase across the three time trend groups? This is likely a significant driver of the prevalence of comorbidities and AICC.
Was the time (2000-2005, 2006-2010, 2011-2015) determined based on the time of diagnosis into the study? If a patient was enrolled in 2004 and followed up until 2010, that patient was not counted into the 2006-2010 group?
“Disease-free survival was calculated from the date of surgery to the first record of metastasis, local recurrence or death.”
Comment: Is this death from colorectal cancer or all-cause mortality?
“Only variables with P < 0.20 were entered in the multivariate analysis.”
How did the authors decide to use <0.20?
The rate of multimodal treatment increased over time but for patients with stage I-III colon and rectal cancer overall and cancer-specific survival remained stable over time. And for stage IV patients, OS and CSS improved significantly throughout the study period, particularly for rectal cancer. The authors should explain if multimodal treatment increased over time by cancer stage and how multimodal treatment relates to survival by cancer stage.
Reviewer 2 Report
The authors conducted a retrospective analysis of all CRC that were resected at their Institution from 2000 to 2015. They stated they wanted to assess how multimodal management changed over time, and the impact of age and comorbidities on treatment and survival. They highlighted an increased use of laparoscopy, adjuvant chemotherapy, and multimodal treatments for metastatic disease, which are in line with current literature. Also, they stated that patients with high ACCI received neo-adjuvant CRT for rectal cancer less frequently, as well as adjuvant chemotherapy and surgical resection for metastatic disease. Finally, they showed that ACCI significantly influenced OS, while it did not affect CSS or DFS.
In my opinion this study does not add any novel information to current literature. It is well known that laparoscopic surgery and multimodal treatment (both for rectal cancer and oligo-metastatic colon cancer) have become more and more common, and they are currently suggested from all international guidelines. Moreover, ACCI has been developed to predict the ten-year mortality for a patient based on his age and comorbidities. Its association with overall survival (all-causes mortality) is therefore obvious, as it is its irrelevance in predicting cancer-specific and disease-free survival. I also question the choice of excluding present colorectal cancer from ACCI calculation, since the presence of metastatic disease (which adds 6 points on ACCI score) may have changed significantly the patient's comorbidity burden. Finally, I don't agree with the conclusion presented both in the abstract and the discussion paragraph. The authors state that "Comprehensive geriatric assessment and prehabilitation program might improve the management of older patients in the future." However, most of the comorbid conditions considered towards calculation of ACCI are not modifiable (age, myocardial infarction, TIA, COPD, dementia, lymphoma, ecc), thus not amenable of improvements through prehabilitation programs.
Round 2
Reviewer 1 Report
The authors have satisfactory changes. I have no further comments.
Author Response
No comments to respond to.
Reviewer 2 Report
I thank the authors for the changes they have made to the original paper, which I think has now improved in terms of quality and interest. They focus on the impact of age and comorbidity on management and outcomes after surgery for colorectal cancer, and they find that patients with high ACCI are less frequently submitted to multimodal treatment and major resections with poorer overall survival. However, cancer-specific survival does not differ between ACCI groups.
Although the quality of the paper has improved, I still have some minor comments:
- Page 7, lines 345-351: I appreciate the changes the authors made to the figure. However, the text is quite confusing since it is a list with a lot of punctuation, parentheses, and percentages that is difficult to read.
- Page 7, lines 349-351: the authors reports data on CSS and DFS but there is no reference to a Figure or a Table. If those data are not reported anywhere else, I would describe them more in details in the text.
- Page 7, lines 352-354: the author does not state the variable for which multivariate analysis was conducted (OS? CSS? DFS?).
- Discussion, page 8-9, lines 466-484: the author state that ACCI negatively influence the use of neo-adjuvant chemo(radio)therapy, adjuvant chemotherapy, post-operative mortality and morbidity management of metastatic disease and overall survival. However, they found no differences in terms of CSS or DFS. I think that this point should be emphasized: it seems that patients with high ACCI (either due to comorbity burden or age) present similar oncological results despite they are treated less aggressively in terms of chemotherapy and multimodal treatment.
- Page 9, lines 492-503: I would delete the whole paragraph, since these data are only marginally reported in the Result section and Supplementary materiels. Moreover the fact that laparoscopy use increased over time is well-known and those not add any relevant information.
- After the drawbacks and strenghts paragraph, I would like to read a conclusion statement.
- Lines are not numbered correctly since there are some missing.
Author Response
I thank the authors for the changes they have made to the original paper, which I think has now improved in terms of quality and interest. They focus on the impact of age and comorbidity on management and outcomes after surgery for colorectal cancer, and they find that patients with high ACCI are less frequently submitted to multimodal treatment and major resections with poorer overall survival. However, cancer-specific survival does not differ between ACCI groups.
Although the quality of the paper has improved, I still have some minor comments:
- Page 7, lines 345-351: I appreciate the changes the authors made to the figure. However, the text is quite confusing since it is a list with a lot of punctuation, parentheses, and percentages that is difficult to read.
A: We agree that the data would have been better presented in a figure, as was originally done. However, removal of some of them was requested by both reviewers earlier, and consequently, the data need to be presented. We are also comfortable presenting the data without confidence intervals so the paragraph (p7, 225-232) has been modified as follows:
“Five-year OS from stage I-IV colon and rectal carcinoma was worse in the high ACCI group compared to low and medium ACCI groups: For colon (Fig.2A), the OS was 71.5% for low ACCI, 71.1% for intermediate ACCI and 49.8% for high ACCI. For rectum (Fig. 2B) the OS was 74.4%, 67.8% and 60.4%, respectively. Five-year CSS by low, intermediate and high ACCI from stage I-IV colon was 75.9%, 77.2% and 68.9% (p=0.056) and from rectal cancer 78.3%, 79.1% and 79.4% (p=0.913), respectively. Five-year DFS by low, intermediate and high ACCI from stage I-III colon cancer was 78.3%, 80.5% and 80.1% (p=0.904) and from stage I-III rectal cancer 75.9%, 74.4 and 76.5 (p=0.788), respectively.”
- Page 7, lines 349-351: the authors reports data on CSS and DFS but there is no reference to a Figure or a Table. If those data are not reported anywhere else, I would describe them more in details in the text.
A: These data were removed as a response to referee #2’s previous critique “[ACCI] is irrelevant in predicting cancer-specific and disease-free survival”. As this comment is directly contradictory, and we agree they are relevant, we have now returned these data to the revised manuscript, and reported the survivals as text (p7, r226-233). However, doing so also contradicts the previous comment #1 wishing to report less data in the same paragraph.
- Page 7, lines 352-354: the author does not state the variable for which multivariate analysis was conducted (OS? CSS? DFS?).
A: Overall survival. This information was already given in the title of table 3, but we have added this also in the text, p7, r236.
- Discussion, page 8-9, lines 466-484: the author state that ACCI negatively influence the use of neo-adjuvant chemo(radio)therapy, adjuvant chemotherapy, post-operative mortality and morbidity management of metastatic disease and overall survival. However, they found no differences in terms of CSS or DFS. I think that this point should be emphasized: it seems that patients with high ACCI (either due to comorbity burden or age) present similar oncological results despite they are treated less aggressively in terms of chemotherapy and multimodal treatment.
A: We agree with this. We have added the following text to p10 r359-361: “It seems that patients with high ACCI present similar oncological results despite they are treated less intensively in terms of chemotherapy and multimodal treatment.”
- Page 9, lines 492-503: I would delete the whole paragraph, since these data are only marginally reported in the Result section and Supplementary materiels. Moreover the fact that laparoscopy use increased over time is well-known and those not add any relevant information.
A: The page and row numbers stated by the referee do not seem to refer to the current manuscript version in the submission system. We think, however, that the referee means removing the following text, as we have now done:
“Level 1 evidence from several large randomized trials comparing laparoscopic versus open resection for colon cancer (COST, Color I, CLASICC) have shown that the laparoscopic method can provide an equivalent oncological outcome, a similar rate of complications, and a faster short-term recovery than the open method [3]. Moreover, Color II, CLASICC and COREAN trials and two recent randomized trials (AlaCart and Z6051) have shown no survival differences from rectal carcinoma between laparoscopic and open resections, suggesting that minimally invasive surgery is oncologically as effective as open surgery for rectal carcinoma [3].”
We have kept the following text:
“Changes in treatment from 2000 to 2015 were seen next to a major increase of laparoscopic surgery, increased use of adjuvant chemotherapy and an intensified treatment of metastatic disease with a shift to an increased use of neoadjuvant chemotherapy and metastasectomy, particularly for rectal carcinoma. Laparoscopic and open resections were evenly distributed between the ACCI groups.”,
as these data are reported in the results, and are relevant for the discussion and conclusions.
- After the drawbacks and strenghts paragraph, I would like to read a conclusion statement.
A: We have added a conclusion statement corresponding the abstract on p10 r387-389: “Despite improvements in multimodal management over time, old age and high comorbidity burden affect the use of adjuvant chemotherapy, preoperative (chemo)radiotherapy and management of metastatic disease, and worsens overall survival from CRC.”
- Lines are not numbered correctly since there are some missing.
A: This seems to be a manuscript handling problem related to editorial process as a Word format on different versions of MS Word. We have not deliberately added nor edited the row numbers, but agree that this causes problems for understanding which part of the text the referee means. As far as we see, no row numbers have been presented twice. The line numbers are usually not printed to the publication so the problem is solved by copyediting and removing them.